

# Diversity of edible insects in a Natural World Heritage Site of India: entomophagy attitudes and implications for food security in the region

Jayanta Kr Das[1], Arup Kumar Hazarika[2], Unmilan Kalita[3], Subhash Khanna[4], Tarali Kalita[2] and Sangeeta Choudhury[2]

[1] Department of Zoology, Barama College, Barama, Assam, India
[2] Department of Zoology, Cotton University, Guwahati, India
[3] Department of Economics, Cotton University, Guwahati, India
[4] Swagat Super Speciality Surgical Hospital and Swagat Academy of Medical Sciences, Guwahati, India

Corrected 21 December 2023

Corresponding authors
Arup Kumar Hazarika,
dr.arupkhazarika@gmail.com
Unmilan Kalita,
unmilan.k@gmail.com

## ABSTRACT

Insects not only play a significant role in the ecological process of nature but since pre-historic times have also formed a part of the human diet. With a still growing population and skewed demographic structures across most societies of the world, their role as nutrient-rich food has been increasingly advocated by researchers and policymakers globally. In this study, we examine the edible insect diversity and entomophagy attitudes of ethnic people in Manas National Park, a UNESCO Natural World Heritage Site, located in Assam (India). The study involved a field investigation through which the pattern of entomophagy and the attitude towards insect-eating was studied. Following this, we examined the edible insect diversity and abundance at different sampling points. A total of 22 species of edible insects belonging to fifteen families and eight orders were recorded from different habitat types. Out of these 22 species, Orthopterans showed a maximum number of eight species followed by Hymenoptera (four), Hemiptera (three), Lepidoptera (two), Blattodea (two) and one species each from Coleoptera, Odonata, and Mantodea. Dominance, diversity, and equitability indices were computed along with the relative abundance of the insects concerning four habitat types. Aspects of the economic significance of entomophagy were also observed during the field investigation. To manage insects in the interest of food security, more attention should be given to sustainable collecting and rearing methods emphasizing their economic, nutritional, and ecological advantages.

## INTRODUCTION

Insects are the most diverse and abundant forms of life and constitute a primary component of the total faunal biodiversity on Earth. They play vital roles in an ecosystem that includes soil turning and aeration, dung burial, pest control, pollination, and wildlife nutrition (*Bernard & Womeni, 2017*). Besides providing ecological services, insects are also an

important source of protein, fat, carbohydrate, and other nutrients. As per the current scientific literature, there are 1.4 million species of insects worldwide which are an intrinsic part of the Earth's ecosystem. As such, they arouse interest not only with their immense species richness but also with their species variety and their role in energy flow. A dimension of their existence not to be overlooked pertains to the fact that they have formed a part of human diets since prehistoric times. Evidence points to at least 113 countries where insects form or formed a part of human diets in one way or the other. This practice of consuming insects as part of the human diet is referred to as entomophagy (*Evans et al., 2015*). Insect-eating or entomophagy is nowadays no longer a traditional or common practice in most countries, except for some in South- and South-East Asia, Latin America, and Africa (*Rumpold & Schluter, 2013*), where more than 2,000 insect species are consumed (*Jongema, 2015*). Given the shortfalls of the 'green revolution' and high risk of food insecurity in developing and underdeveloped nations, the use of insects as a potential source of food for the burgeoning human population had been advocated by *Meyer-Rochow (1975)*, a suggestion that has been gaining interest among researchers, entrepreneurs and policy makers worldwide ever since.

Insect farming is popular in many Asian nations for food, feed, and other purposes (*Zhang, Tang & Cheng, 2008*). Weaver ants (*Occophylla smaragdina*), whose chemical composition and value as a human food item is well known are widespread in the Asia-Pacific region and are found from China's south to northern Australia and as far west as India. Although edible insects are not yet of much commercial value, some economic and marketing data on edible insects in Asia and the Pacific are available scarce (*Johnson, 2010*). Approximately 50 insect species are eaten in Thailand's north and about 14 species are eaten by people in southern Thailand (*Rattanapan, 2000*).The insect-eating habits in various regions depend on the indigenous populations' cultural practices, religion and the place they call home. But insects used as emergency food during natural calamities or other national contingencies as well as for their organoleptic characteristics can also be important (*Dumont, 1987*).

The North–Eastern part of India has diverse ethnic groups that have a unique culture of food intake with insect-eating mostly prevalent amongst rural tribal people of the region which have a long-cultured history. A total of 81 species are eaten in Arunachal Pradesh by the Galo and Nyishi tribes (*Chakravorty, Ghosh & Meyer-Rochow, 2011*). Odonata were consumed the most followed by Orthoptera, Hemiptera, Hymenoptera, and Coleoptera.

Scientific reports indicate insects to be significant sources of not only proteins and vitamins, but also lipids, minerals, fibre and carbohydrates. Insects possess a viability of providing daily requirements of these nutrients in most developing countries (*Bukkens, 1997*; *Elemo et al., 2011*). For instance, edible aquatic beetles play an important role in the nutrition and economy of the rural population in Asian, Latin American and African nations (*Macadam & Stockan, 2017*) and are popular in Manipur (*Shantibala, Lokeshwari & Debaraj, 2014*). It should be noted that the diversity and abundance of insects in different habitat types have an observed correlation with the entomophagy attitude of a particular region. Therefore, research indicates the importance of exploiting insect diversity effectively

through insect farming to avoid global problems associated with dependency on a limited number of insect species as experienced with some food animals and crops.

In this research article, we have made an effort to study the edible insect diversity of a UNESCO Natural World Heritage Site, located in the Indo-Burmese biodiversity hotspot. Regional entomophagy was studied through a field investigation. We made an effort to determine the degree to which the ethnic people use insects in their diet and which species they consume. Recording seasonal abundance and availability of edible species as well as evaluating the role that entomophagy could possibly play as a measure of food security in the region, were further aspects of this study.

## MATERIALS & METHODS

### Study Area

The Manas National Park (MNP), located at 26.6594°N, 91.0011°E, was declared a UNESCO Natural World Heritage Site (WHS) in 1985 (Fig. 1). Renowned for its array of rich, rare, and endangered wildlife not found anywhere else in the world, the faunal diversity of MNP includes the Pygmy Hog, Golden Langur, Hispid Hare, Assam roofed turtle and so on. Located at the Himalayan foothills of India, MNP is shares land territory with Bhutan where it is known as the Royal Manas National Park. The park is composed majorly of grassland and a forest biome. It is covered by the Brahmaputra Valley semi-evergreen forest vegetation along with the Himalayan subtropical broadleaf forests and the Assam Valley semi-evergreen alluvial grassland vegetation. This renders MNP a region of rich and abundant biodiversity. Major trees include the *Bombax ceibar, Gmelina arborea, Bauhini purpurea, Syzygium cumin, Aphanamixis polystachya, Oroxylum indcum,* etc. The climate is sub-tropical with a warm and humid summer, followed by a cool and dry winter. Temperatures range from 10 °C to 32 °C.

The park has more than 58 fringe villages directly or indirectly dependent upon it, distributed across three ranges: Bansibari, Bhuiyaparaa and Panbari. The village Agrang lies at MNP's core while most are located in its buffer zone. Spread over the State of Assam's Barpeta and Bongaigaon districts, the tribal population in its fringe areas predominantly include Bodos and Rabhas among which the practice of insect eating and rearing are widespread.

### Insect sampling

Insects were collected using entomological nets, beating tray, water traps, or through digging and handpicking. The local people of the study area helped in the collection process. Insects were usually collected during the early hours of the day (0500–0900 h).

The flying insects were collected via entomological nets at a time when they were active (mid-morning/late afternoon). Sweep nets were used for collecting grasshoppers and other insects which hid in low grass- or herb-dominated vegetation and in small shrubs. Netting was normally carried out during early hours of the day as we could not collect nocturnal taxa in this way. In order to catch nocturnal species, we used light traps. Nocturnal arthropods like species of moths and beetles are easily attracted towards artificial light sources. Light traps have therefore been widely used in nocturnal insect sampling. A high-power CFL

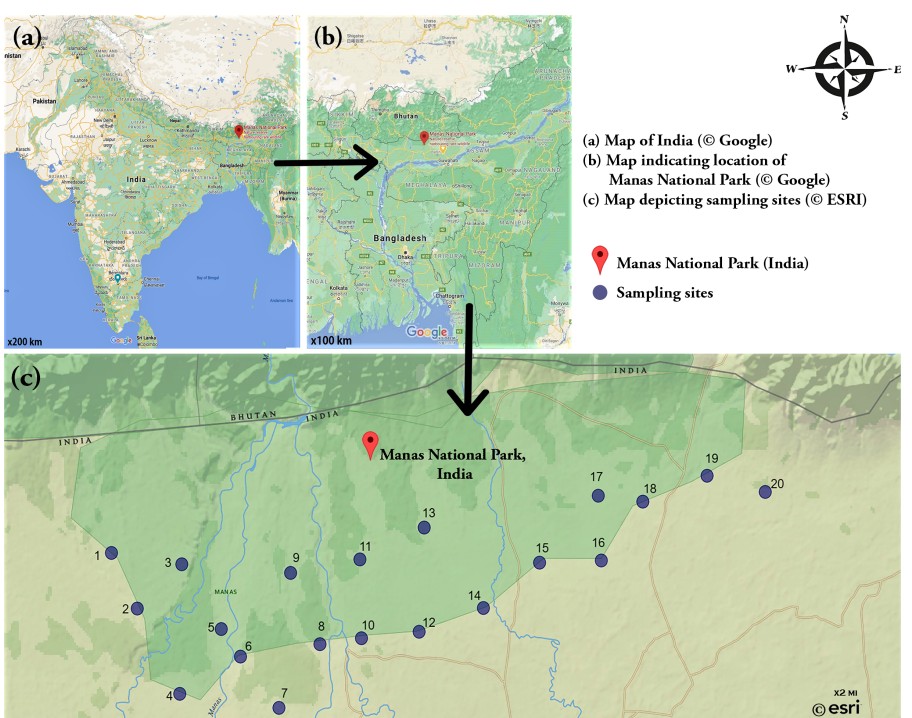

**Figure 1** **Study Area.** (A) Map of India (© Google) (B) Map indicating location of Manas National Park (© Google) (C) Map depicting sampling sites (© ESRI).

bulb was arranged in front of a white cloth for trapping nocturnal insects. Generally, a bowl filled with water was placed under the light sources in the evening, after rainfall, to attract termites. The light attracted the reproductive termites which came out for nuptial flights and were trapped in the water or collected by hand from the water to prevent them from escaping. Light trapping was used widely in case of agricultural habitat type and open field habitat type.

Beating trays were used to collect insects such as Lepidoptera and Hymenoptera. Shrubs and small trees were sampled through commonly used beating tray sample method. Moreover, the red weaver ants were harvested by plucking the nest from the tree and dropping it in a bucket of water before being sorted out for consumption. The soil dwelling edible insects were collected by digging with the help of spades. Besides sweep netting, large insects such as grasshoppers and beetles were also collected by hand which were caught early in the morning or in the evening when they were less mobile due to their low body temperature. Mole and field crickets were dug out of holes.

We used the hand-netting technique to collect the aquatic insects along with other local traditional equipment like *Jakoi*, *Chaloni*, etc. The *Jakoi* is a species of wicker work shovel that is either dragged along the bottom or placed on the water bed to catch the aquatic insects which take refuge in it when the weed is trampled. It is prepared with bamboo slips, which are locally known as 'dai'. 'Jati' bamboo is specially used for making this particular implement. *Chaloni* is a bamboo strainer used to separate insects from collected water.
Long handled aquatic net was used to collect insects that live on the water surface. Many adult insects living on the surface were predators, so they were removed from the net using forceps directly into a collection container. The kick-net method which is a process where insects are collected by dislodging them from the substrate (habitat) was also used. The organisms that were dislodged by the disturbance were collected on the net.

For preservation of specimens, both dry and wet preservation methods were followed. For dry preservation, the specimens were preserved using pins in insect cabinet box and were mainly sun-dried. Soft-bodied insects were preserved using 70% ethyl alcohol. Besides, some hard-bodied edible insects were preserved using 2–3% formaldehyde (*Ghosh & Sengupta, 1982*). Identification was done later by comparison with other specimens. Some were identified in the Zoological Survey of India, Shillong, Meghalaya (India).

Sampling was done from 20 chosen sites located around MNP during the period 2018 (June)–2019 (June). The permission for conducting the field study was obtained from Office of the Principal Chief Conservator of Forests (Wildlife) and Chief Wildlife Warden, Government of Assam, India vide No. WL/FG31/ResearchStudyPermission/19th Meeting/2019. The remaining methodology of the study is outlined in Fig. 2.

## Edible insect density, diversity and abundance

Studying the diversity required us to divide each sampling point into four different habitat types, namely, open field habitat (OFH), forest/backyard forest habitat (FBH), swampy area habitat (SAH), and agricultural field habitat (AFH). The entire sampling area amounted to approximately 842 km². Insects were recorded within quadrates (2 m ×2 m dimension) established in the habitat type and monitored for four seasons, namely, pre-monsoon (March, April and May), monsoon (June. July, August and September), retreating monsoon (October and November), and winter (December, January and February) (*Borthakur, 1986*).

The Shannon–Wiener index (H') for diversity, Simpson index (D) for dominance, and Margalef index for species richness in the four selected habitat types were computed. Order-wise relative abundance and species-wise abundance in the different habitats were also computed. The descriptions and mathematical expressions are outlined below. The indices were estimated using PAST (v.3.26) (*Hammer, Harper & Ryan, 2019*) and SPSS (v.23).

Shannon–Weiner index (H′) determines the diversity of insect species in a particular habitat type. The higher the H' value, the greater is the diversity. Expression (i) gives the formula.

$$\mathrm{H}' = -\sum p_i ln p_i \ldots \ldots \tag{1}$$

where $p_i$ = proportion of individuals found in $i^{th}$ species

Simpson's index (D) defines the probability of drawing any two individuals at random from a very large community of the same species. If D increases, we can say that diversity has decreased. This index, defined by expression (ii), accounts for both aspects of diversity, i.e., richness and evenness.

$$\mathrm{D} = \sum \left( \frac{\sum n_i [n_i - 1]}{N[N-1]} \right) \ldots \ldots \tag{2}$$

where, $n_i$ = individuals in $i^{th}$ species, N = total number of individuals

Margalef's index (R) gives a precise idea about a species' richness. It attempts to compensate for the effects of sampling by taking a ratio of species richness by the total number of individuals in a sample, given in expression (iii).

$$R = (S - 1)/\ln N \tag{3}$$

where, S = total species in a community, N = total number of individuals in that community.

### Entomophagy study

Understanding the entomophagy attitudes and distribution among the tribal population near MNP required conducting a survey. Methods included interactions with the villagers through questionnaires, field surveys, and a market survey. The villages were selected randomly and were surveyed once per season for the whole year. Questions were asked to a mixed group of ethnic people which included individuals from all sections of the society. The market survey helped record the economic importance of these insects for the local economy. Questions pertained to the number of insects sold per week/month, their market prices, and how popular were the insects in ethnic cuisine. Overall, the questionnaire survey included 2,672 respondents from 30 villages of which 981 were from the Adivashi tribe, 695 were Bodos, 436 were Saranias, 422 were Rabhas and 138 were non-tribal individuals. Written consent was obtained from the respondents during the field interviews.

## RESULTS

Table 1 shows the order-wise number of edible insects found in the study area. In MNP, the order Orthopteran recorded the maximum number with 8 species, followed by Hymenoptera with 4 species. The order Hemiptera was found to have 3 species followed by Lepidoptera and Blattodea with 2 species each. The order Coleoptera, Mantodea, and Odonata accounted for 1 from each species and family. A total of 5,614 edible insects were recorded from AFH, 1,318 in FBH, 3,306 in OFH and 2,096 individuals in SAH, during the field observation. No common abundant species was found in a single habitat. Most of the insects were found in two or three habitats during the study period.

Table 2 showcases the types of edible insects consumed by the ethnic people. In this table, the local and common name (in Bodo), the scientific name with their taxonomy, and their seasonal availability, edible parts, and mode of consumption are tabulated. Common names in Bodo have been displayed in Table 2 as they were more popular among the local people. Seasonal availability was maximum during June to September, gradually reducing towards the winter season. Species of the order Orthoptera were most abundant in May to September, whereas, Coleopterans were usually available from April to September. Insects belonging to the Hemiptera and Hymenoptera were found to be restricted to the period lasting from April to October, whereas, Mantodea were available from June to October. Some edible insects like *Lethocerus indicus, Periplaneta americana* and *Gryllotalpa africana* were found to be available throughout the year, but in the winter, they were less abundant than during the pre-monsoon and monsoon season.

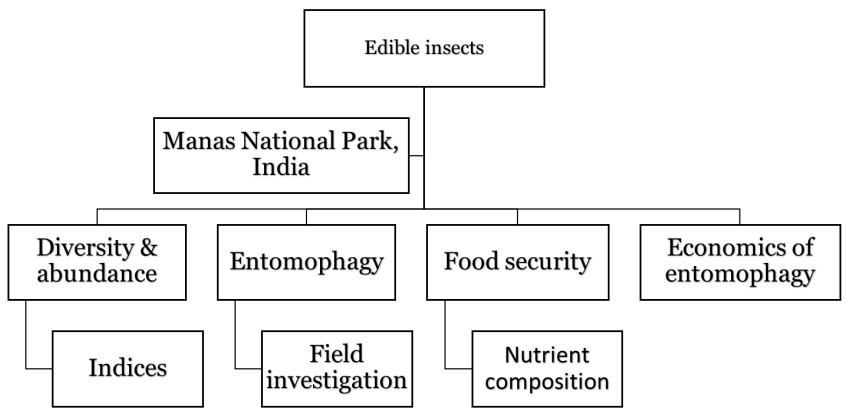

**Figure 2** **Methodology.**

**Table 1** **Order-wise number of edible insects.** Each data point in the right column indicates the number of insects present in the species-type specified in the left column.

| Order | Number of species |
| --- | --- |
| Coleoptera | 5 |
| Hemiptera | 3 |
| Hymenoptera | 4 |
| Lepidoptera | 2 |
| Blattodea | 2 |
| Odonota | 1 |
| Montodea | 1 |
| Orthoptera | 8 |
| Total | 26 |

Simpson index (D) for dominance, Shannon–Wiener index (H′) for diversity, and Margalef index for evenness/equitability were calculated in the four selected habitats (Table 3). Further, species abundance was found to be the highest in *Chondracris rosea* with 17.64, followed by *Philosamia ricini* with 6.50 in AFH. In FBH, the highest species abundance was found in *Microtermes obesi* with 6.00, followed by *Periplaneta Americana* with 4.00. In OFH, *Gryllus bimaculatus* was the highest abundant species with 5.1, followed by *Lethocerus indicus* with 4.17. Table 4 shows the relative abundance of the edible species in selected habitats. *Chondracris rosea* has the highest relative abundance (11.50%) followed by *Choroedocus robustus* (8.92%), the least relative abundant insect species includes *Laccotrephes ruber* (0.42%).

Further, the proportion of ethnic communities practicing entomophagy in MNP has been graphically represented in Fig. 3. As mentioned before, the 2,672 respondents to our survey included 981 individuals from the Adivashi tribe, 695 from the Bodo tribe, 436 from the Sarania tribe, 422 from the Rabha tribe and a total of 138 individuals were non-tribal. We also categorised the respondents of our survey who considered the insect-eating habit favourable, into four age-groups, namely, less than 60 years, between 40-60 years, between

Das et al. (2020), *PeerJ*, DOI 10.7717/peerj.10248

**Table 2** **Taxonomy with seasonal availability of edible insects in MNP.** Each data point indicates the scientific name, order, family, English name, local name, seasonal availability, edible part, and mode of eating of a particular edible insect.

| Scientific name | Order | Family | English name | Vernacular name (Bodo) | Seasonal availability | Edible part | Mode of eating |
|---|---|---|---|---|---|---|---|
| *Eretes stictus* | Coleoptera | Dytiscidae | Larva of diving bettle | Jujema | Whole year | Larvae | Fried |
| *Hydrophilus olivaceus* | Orthoptera | Hydrophilidae | Water Scavenger | Ankhaouri | Whole year | Larvae and Adult | Fried or Curry |
| *Orycetes rhinoceros* | Coleoptera | Scarabaeidae | Rhinoceros beetle | Jeljer | Sep-Feb | Larvae (Grubs) | Fried |
| *Phyllophaga spp.* | Coleoptera | Scarabaeidae | June beetle | Bwarbi | April- June | Adult | Fried |
| *Plectroderma scalator* | Coleoptera | Cerambycidae | Wood borer | GalaGunjer | May- August | Larvae | Fried |
| *Diplonychus rusticus* | Hemiptera | Belostomatidae | Water beetle | AmphuDabla | May-Sep | Adult | Fried or Curry |
| *Laccotrephes ruber* | Hemiptera | Nepidae | Water scorpion | Omabunda | Jun-Oct | Adult | Fried or Smoked |
| *Lethocerus indicus* | Hemiptera | Belostomatidae | Giant Water bug | Gangjema | Whole Year | Adult | Fried or Smoked |
| *Apis dorsata* | Hymenoptera | Apidae | Rock bee | Berema | May-Sep | Eggs & Larvae | Raw |
| *Parapolybia varia* | Hymenoptera | Vespidae | Lesser paper wasp | Mwsousalai bere | April-Oct | Larvae | Fried, Raw |
| *Polistis olivaceus* | Hymenoptera | Vespidae | Paper wasp | Jothabere | April-Oct | Eggs & larvae | Raw, Fried, Smoked |
| *Vespa affinis* | Hymenoptera | Vespidae | Potter wasp | Handilore bere | April- September | Eggs & larvae | Raw, Roasted, Fried |
| *Mantis religiosa* | Mantodea | Mantidae | Praying mantis | Gumagangu | June-November | Adult | Fried, Smoked |
| *Ictinogomphus rapax* | Odonota | Gomphidae | Dragon fly | Gandula | March- August | Nymph | Fried |
| *Acheta domestica* | Orthoptera | Gryllidae | House Cricket | Gusengra | May-Sep | Adult | Fried, Smoked |
| *Chondracris rosea* | Orthoptera | Acrididae | Short-horned Grasshopper | Gumanarenga | June-August | Adult | Fried |
| *Choroedocus robustus* | Orthoptera | Acrididae | Short-horned Grasshopper | Gumakhushep | June-Oct | Adult | Fried |
| *Eupreponotus inflatus* | Orthoptera | Acrididae | Short-horned Grasshopper | Gumanargi | May-Sep | Adult | Fried or Smoked |
| *Mecopoda elongate* | Orthoptera | Tettigoniidae | Long-horned grasshopper | Gumakhufri | May-Sept | Adult | Roasted or Fried |
| *Gryllotalpa africana* | Orthoptera | Gryllotalpidae | Mole cricket | Sosroma | Whole Year | Adult | Fried or Smoked |
| *Gryllus bimculatus* | Orthoptera | Gryllidae | Field Cricket | Fendadangra | May-Sep | Adult | Fried, Smoked |
| *Heiroglyphus banian* | Orthoptera | Acrididae | Grasshopper | Gumagudul | June-Oct | Adult | Fried, Smoked |
| *Periplaneta americana* | Blattodea | Blattellidae | Cockroach | Thaoamphow | Whole year | Adult | Fried |
| *Microtermes obesi* | Blattodea | Termitidae | Termite | Wuri | March- July | Larvae, Adult | Fried |
| *Philosamia ricini* | Lepidoptera | Saturnidae | Erisilkworm | Amphoulata | April-Sept | Larvae | pupae fried |
| *Antheraea assama* | Lepidoptera | Saturnidae | Muga silkworm | Amphumuga | April-Sept | Larvae, pupae | Fried |

**Table 3 Diversity indices (habitat type) of edible insects recovered from four selected habitats.** Each data points indicate the different diversity indices of a particular insect type with respect to different habitat types.

| | AFH | FBH | SAH | OFH |
|---|---|---|---|---|
| Species Richness | 24 | 22 | 6 | 23 |
| Total individuals encountered | 9213 | 1455 | 3435 | 6497 |
| Simpson | 0.1148 | 0.3871 | 0.2423 | 0.1467 |
| Shannon-Wiener | 2.822 | 2.153 | 1.329 | 2.392 |
| Margalef | 2.936 | 1.836 | 0.653 | 2.294 |

Notes.
AFH, Agricultural field habitat; FBH, Forest/backyard forest habitat; SAH, Swampy area habitat; OFH, Open field habitat.

20,-40 years and greater than 20 years (Fig. 4). Consumers in the 20-,40 group responded highly favourably while those in less than 20 years group responded less favourably owing to different variations of entomophobia. There are various reasons for eating insects which were found among the different ethnic groups during the questionnaire survey (Fig. 5). The different modes of insect consumption have been presented in Fig. 6.

## DISCUSSION

### Edible insect diversity and abundance

As part of this study, we find that species of the order Orthoptera are popular among the ethnic people for consumption purposes. The edible species majorly include both short and long-horned grasshoppers (*Eupreponotus inflatus, Choroedocus robustus, Chondracris rosea, Mecopoda elongata and Hieroglyphus banian*), field crickets (*Gryllus bimculatus*), house crickets (*Acheta domesticus*) and mole crickets (*Gryllotalpa Africana*). Other species include potter wasp (*Vespa affinis)* and paper wasp (*Polistis olivaceus*), Indian honey bee (*Apis indica*) and rock bee (*Apis dorsata*), giant water bug (*Lethocerus indicus*) and some others. The ethnic (tribal) communities consuming these insects were mainly those of the Adivashis, followed by the Bodo, Rabha, and Sarania. A section of the non-tribal population also consumed insects as part of their diets.

Species diversity, richness, and evenness gives an idea about the variety and diversity of species in the study sites. The most commonly used dominance and diversity indices in ecology are the Simpson index and the Shannon–Wiener index. Simpson index is used to assess the dominance but fails to provide an idea about species richness. Shannon–Wiener index is expected to determine both diversity characteristics (evenness and richness) but does not provide any information on rare species which, however, are very important in studies of biodiversity. Our results show that the species dominance is highest in FBH (0.3871), followed by SAH (0.2423), OFH (0.1467), and AFH (0.1148). On the other hand, species diversity, as per H', was highest in AFH (2.822), OFH (2.392), FBH (2.153) and SAH (1.329). This establishes the fact that as insect diversity decreases, their dominance should increase. In MNP, this can be noticed for the forest habitat. Further, this result is corroborated by the Margalef index which is found to be highest for AFH (2.936), OFH (2.294), FBH (1.836), and SAH (0.653).

**Table 4  Abundance of edible insect in three different terrestrial habitats.** Each data point shows the abundance of different edible insects in the terrestrial habitat types chosen in our study.

| Order | Species | AFH | Quadrate occurrence | Abundance | FBH | Quadrate occurrence | Abundance | OFH | Quadrate occurrence | Abundance |
|---|---|---|---|---|---|---|---|---|---|---|
| Orthoptera | *Eupreponotus inflatus* | 1,205 | 212 | 5.68 | 0 | 0 | 0.00 | 73 | 32 | 2.28 |
| Orthoptera | *Mecopoda elongata* | 12 | 9 | 1.33 | 44 | 11 | 4.00 | 5 | 4 | 1.25 |
| Orthoptera | *Choroedocus robustus* | 1,256 | 206 | 6.10 | 8 | 7 | 1.14 | 40 | 27 | 1.48 |
| Orthoptera | *Hieroglyphus banian* | 1,224 | 212 | 5.77 | 0 | 0 | 0.00 | 66 | 21 | 3.14 |
| Orthoptera | *Gryllus bimaculatus* | 41 | 8 | 5.13 | 79 | 45 | 1.76 | 1,043 | 208 | 5.01 |
| Orthoptera | *Acheta domesticus* | 251 | 71 | 3.54 | 155 | 59 | 2.63 | 4445 | 148 | 3.01 |
| Orthoptera | *Gryllotalpa africana* | 56 | 48 | 1.17 | 29 | 16 | 1.81 | 532 | 153 | 3.48 |
| Orthoptera | *Chondracris rosea* | 988 | 56 | 17.64 | 24 | 11 | 2.18 | 76 | 44 | 1.73 |
| Hymenoptera | *Vespa affinis* | 4 | 1 | 4.00 | 178 | 72 | 2.47 | 3 | 1 | 3.00 |
| Hymenoptera | *Polistis olivaceus* | 43 | 36 | 1.19 | 189 | 49 | 3.86 | 44 | 42 | 1.05 |
| Hymenoptera | *Apis indica* | 0 | 0 | 0.00 | 0 | 0 | 0.00 | 44 | 27 | 1.63 |
| Hymenoptera | *Apis dorsata* | 0 | 0 | 0.00 | 110 | 76 | 1.45 | 28 | 8 | 3.50 |
| Hemiptera | *Lethocerus indicus* | 58 | 23 | 2.52 | 6 | 3 | 2.00 | 25 | 6 | 4.17 |
| Hemiptera | *Laccotrephes ruber* | 24 | 17 | 1.41 | 3 | 2 | 1.50 | 4 | 2 | 2.00 |
| Hemiptera | *Diplonychus rusticus* | 11 | 8 | 1.38 | 5 | 3 | 1.67 | 0 | 0 | 0.00 |
| Lepidoptera | *Antheraea assama* | 2 | 1 | 2.00 | 7 | 7 | 1.00 | 88 | 46 | 1.91 |
| Lepidoptera | *Philosamia ricini* | 13 | 2 | 6.50 | 87 | 49 | 1.78 | 35 | 13 | 2.69 |
| Mantodea | *Mantis religiosa* | 112 | 45 | 2.49 | 28 | 21 | 1.33 | 74 | 60 | 1.23 |
| Blattodea | *Periplaneta americana* | 4 | 3 | 1.33 | 212 | 128 | 1.66 | 384 | 152 | 2.53 |
| Coleoptera | *Oryctes rhinoceros* | 67 | 39 | 1.72 | 20 | 9 | 2.22 | 41 | 11 | 3.73 |
| Blattodea | *Microtermes obesi* | 31 | 21 | 1.48 | 78 | 13 | 6.00 | 72 | 24 | 3.00 |
| Odonota | *Ictinogomphus rapax* | 212 | 67 | 3.16 | 56 | 29 | 1.93 | 184 | 89 | 2.07 |
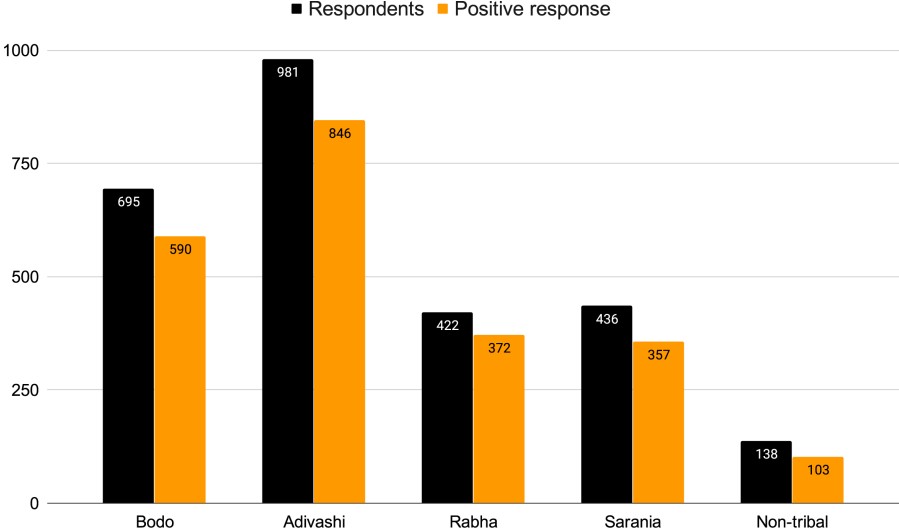

**Figure 3 Entomophagy of different ethnic groups.** The black column indicates the respondent groups. The yellow column indicates the quantum of positive response. The numbers inside the columns indicates the number of people.

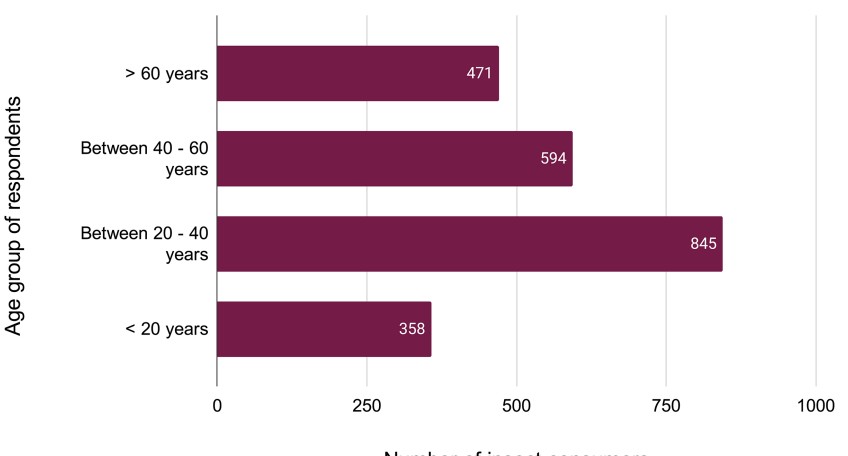

**Figure 4 Age group of respondents favouring entomophagy.**

Notably, forest habitats are the prime source of edible insects for local people. This adverse finding in the case of FBH may be attributed to various reasons. Decreasing forest cover, changes in vegetation type, adverse climatic conditions, or indiscriminate collection and consumption of edible insect. These directly affect the insect diversity and rejuvenation of insect species. In the case of MNP, high temperatures, inadequate rainfall, and vegetation cover may also have influenced the population density of these edible insects. Notably, the overall climate of Assam has warmed by over 0.5 °C for the past decade which is expected to rise up to 2.2 °C by 2050.
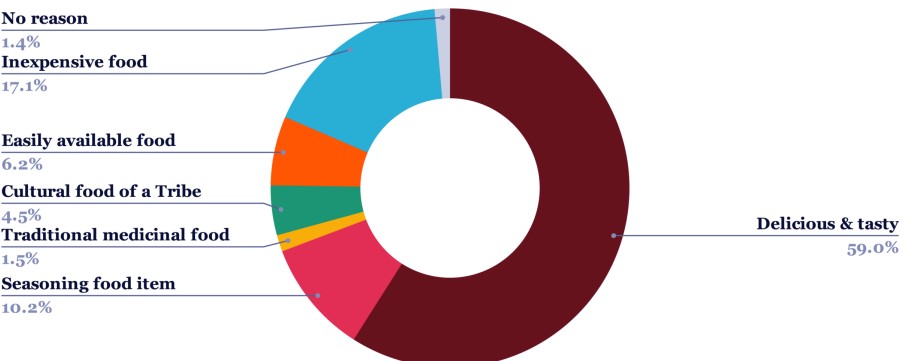

**Figure 5 Different reasons for practicing entomophagy.** The coloured sections of the pie display the different reasons why insect-eating (entomophagy) is practiced by the local people.

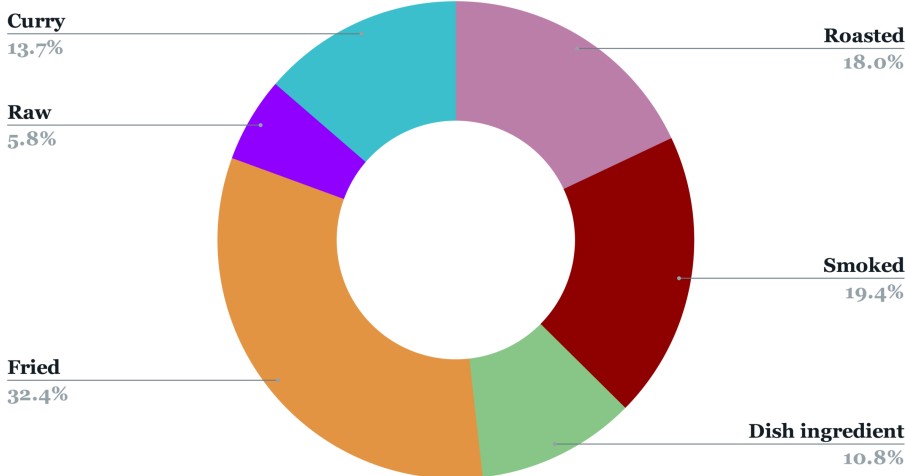

**Figure 6 Different modes of eating insects.** The different coloured sections show the different modes/ways of eating insects by the local people.

It should be noted that Shannon–Weiner and Simpson diversities increase as richness increases for a given pattern of evenness, and increase as evenness increases for a given richness, but they do not always follow the same trend. Simpson diversity is less susceptible to richness and sensitive to evenness than Shannon index which, in turn, is more receptive to evenness. At the other extreme, the Berger-Parker index, depends entirely on evenness- it is simply the inverse of the proportion of individuals in the community that belongs to the single most common species, while the other indices (Margalef) are dependent on the number of species. Apart from the diversity and distribution patterns for insect taxa, interactions between insect groupings and plant groups are another important topic requiring urgent research attention. This is because plants provide key habitat parameters for many insect species ranging from shelter to breeding sites. This has not been covered under this study and could be pointed out as its limitation.

Our analysis of seasonal diversity of edible insect species shows that the diversity of the edible insects was greater during monsoon and pre-monsoon season, moderate in the retreating monsoon season, and lowest in the winter season. As per the survey report, it was found that the abundance of insects found today is much lower than what it was earlier. The decreasing pattern is corroborated by *Doley & Kalita (2011)*, *Narzary & Sarmah (2015)*, *Das, Hazarika & Khan (2012)*, with slight changes. This establishes that seasonal availability of edible insects is declining with time. Further, anthropogenic disturbances and deforestation are seen rampant in the fringes of MNP. Ground-level evidence glaringly shows that villagers are converting forest lands into agricultural fields. This is an outcome of the burgeoning population of Assam where the human population density is 398 persons per $km^2$ which is way above the global density of 14.7 persons per $km^2$. Such anthropogenic pressure (*Morris, 2010*) is bound to destroy species composition, community structure, and insect diversity. This calls for urgent ecosystem restoration to sustain the distribution pattern and abundance of edible insects.

In the regional context, a study of the diversity of insects consumed by the people in Dhemaji District of Assam revealed that a majority of 14 species of insects were used as food (*Doley & Kalita, 2011*). 40 species of edible insects were recorded in Karbi Anglong District of Assam corroborated by *Hanse & Teron (2012)*. Another study involving the ethnic community of the Bodos, recorded 25 species of local insects, belonging to eight orders and fourteen families which are consumed as food (*Narzary & Sarmah, 2015*).

## Entomophagy, food security, and its economic implications

The field investigation revealed that most of the respondents found insects to be tasty and delicious (59%), while a section found them to be an inexpensive source of food (17.1%). Traditional medicinal food is also one of the reasons why edible insects are collected (*Meyer-Rochow, 2017*). This indicates the substantial preference of insects in the food habits of people and underscores their importance in the allocation of household costs and sustaining food security. This can be corroborated with the findings of *Mozhui, Kakati & Changkija (2017)* for Nagaland, where the ethnic people considered insects as a regular food source, rather than an emergency food item. The local people favoured eating insects mostly by frying, roasting, or smoked. This emphasises the wide variety of ways through which insects may be consumed. However, a low percentage of respondents claimed them to be easily available food as collecting them is rather difficult compared to conventional livestock. This calls for the development of an insect farming industry as well. Further, a large number of respondents in the 20–40 years and 40–60 years age bracket favoured eating insects due to the various reasons as in Fig. 4. Entomophagy, as such, is highly popular among the youth population. However, in Ethiopia young people are less inclined to eat or even taste insects (*Ghosh et al., 2020*).

Besides, the nutritional significance of edible insects has been well established by current scientific literature. It is observed that nutrients vary widely across insect species wherein some are rich in protein and lipids while others are rich in mineral content. *Chen, Feng & Chen (2009)* note that edible insects are rich in protein and fat, but sometimes may lack carbohydrate content. However, insects like bees, honeypot ants, etc., are very rich in

carbohydrates. Besides, *Collavo et al. (2005)* note that the presence of high essential amino acids is a major reason for insects having high-quality protein. Majority of the population near MNP belong to low- or lower-middle-income category people. Their demography is skewed towards ethnic backgrounds and hence, the economy is highly underdeveloped. Rearing livestock and maintaining animal husbandry practices, require a substantial amount of money. The piggery sector is robust in this area. Practicing this requires large amounts of land and also involves substantial capital. However, the nutritional benefits gained from it are not enough to compensate for the effort. Also, insects generally meet the WHO recommendation for amino acid content with nymphs being their most abundant source. Coleoptera has a higher amount of protein than most livestock. More importantly, edible insects bear many non-health related benefits related to environmental and financial costs than livestock.

On the other hand, it is important to note that many edible insects require higher energy in culture and contain higher sodium and saturated fat content (*Payne et al., 2016*). This diminishes their worth as alternative nutrient sources to fight nutrition-related diseases. This is because the saturated fat content of edible insects is not recommended for people with heart disease risk, obesity, or metabolism issues. Further, some beetle or butterfly species produce dangerous toxins that are harmful to human health. Such species must be identified before being consumed as food (*Blum, 1994*). However, insects have very high micronutrient content which can be extracted or consumed at a third of the cost than other food products.

MNP is a highly flood-ravaged area with untimely floods occurring during the sowing period. Floods in 2019 affected over a million people of Assam with a majority from the Baksa District (where MNP is located) and the adjacent district of Barpeta. This frequently uproots the livelihood of the local people rendering them vulnerable to high food insecurity. It should be noted that these ethnic people otherwise have decent livestock and animal husbandry resources. With floods, they tend to lose livestock in a large-scale manner. At this juncture, edible insects can play a significant role in maintaining the nutritional content of their diet intact.

Animal protein is superior to plant; therefore, the best protein supplements should include some animal protein. Thus, insects may provide for good quality protein ingredients to produce a high standard protein supplement for the food industry (*Ssepuuya et al., 2017*). It was also found that the lipid content of common insect larvae (37.87%) are higher than the soybean (14.60%). From the energy point of view, lipids are important because one gram of lipid provides more than 9 kcal of energy when oxidized in the body. Lipids are structural components of all tissues and indispensable in cell membranes structure and cell organelles (*Drin, 2014*). The fat content of pupae and larvae of edible Coleoptera is higher than that of the adult insect. These results coupled with the significant role played by edible insects in the local food habits make it undeniable that the desirability of food security in their context is valid as they can be considered as viable sources of macro- and micro-nutrients for human beings.

Edible insects such as beetles have been a rich source of proteins and also other nutrients for a long time and have been preferred over traditional livestock by several communities all

over the world (*Losey & Vaughan, 2006*). For instance, indigenous communities of Mexico are involved in buying and selling edible insects, which are also processed and sold in urban markets. Insects have low-fat content and as such, there has been a high worldwide demand for edible insects. Additionally, aquatic insects are commonly exported from South Asian nations to the United States which are prepared and served in high-end eateries. The estimated size of this market was approximately USD 40 million in 2015. Moreover, in the Lao PDR, insects can be found in markets as ready-to-eat snacks or fried with lime leaves (*Meyer-Rochow, Nonaka & Boulidam, 2008*). Concerning agriculture, beetles have been found to contribute more than a billion dollars in environmental and economic benefits globally. This comes from the fact that they recycle cattle manure, thereby, improving pasture growth, yielding high agricultural benefits, and thus, augmenting the livelihood of agriculturalists. In the context of MNP, a gap in the literature has been observed wherein comprehensive studies on beetles' economic benefits haven't been witnessed.

Rearing insects can result in environmental benefits with respect to food and feed. Insects can impact organic farming while helping to reduce environmental contamination, as they emit fewer greenhouse gases and ammonia, compared with conventional livestock (*Dangles & Casas, 2019*). Given the inclination of Bodos and other tribes in eating insects and rearing them to an extent, economic policies must target rearing practices of insects, rather than solely focussing on animal husbandry. Therefore, several strategies could be employed that can help in efficiently and sustainably making use of such natural biodiversity in augmenting the societal income and its food security, following learnings of other countries like South Korea (*Meyer-Rochow, Ghosh & Jung, 2019*).

Our study confirms that edible insects are of considerable nutritional value and expanding their acceptability as human food can be expected to improve the nutritional status of people and possibly reduce the insects' costs. With a wider insect diversity, the nutritional status of people should improve while costs get reduced (*Dickie, Miyamoto & Collins, 2019*). For instance, mealworms consist of six fatty acids and unsaturated omega-3 components that are equivalent to those found in commonly consumed fishes, and also higher than those found in pigs and cattle (*Raheem et al., 2019*). Since nutrition has been one of the core components in the evolution of economic policies as well as family welfare, it is necessary that the insect eating habits of ethnic people in the study area must be widely augmented while focussing on the preservation of its insect diversity.

Certain insects like silkworms, honey bees, and as of late bumble bees and wasps have been traditionally domesticated since they are of high economic value. As such, insect farming is much needed in the study area. This concept is widely prevalent in Korea, Thailand, Vietnam, and Laos PDR. Vertical farming is another technique that can strengthen local economics and help exploiting new protein sources (*Specht et al., 2019*). Family-run enterprises are mostly involved in this business along with other firms that have commercialised insects as not only food but also sources of protein and other health supplements.

Insect diversity can be critical for livelihood development since, in some developing countries, the poorest members of a society are engaged in gathering and rearing of mini-livestock (*Mason et al., 2018*). Industrial-scale interventions can also augment their

livelihoods that have now been observed in the case of silkworms of Assam. Given the relatively process of rearing, accessibility, and transportation of insects, the people of the study area can immensely benefit if steps to set up an Insect Marketing Hub, assisted by an Insect Development Authority is set up. The hub should be created following a hub-and-spoke model that would not only pertain to processing and distribution matters but also training and R&D issues.

## CONCLUSIONS

In this study, we recorded edible insect diversity and abundance, characteristics, and attitudes of the ethnic communities involved in entomophagy that are residing in the fringes of the Manas National Park, a Natural World Heritage Site. A total of 22 species of edible insects belonging to fifteen families and eight orders were recorded from different habitat types. Out of these 22 species, we recorded a maximum number of 8 Orthopteran species followed by Hymenoptera (4), Hemiptera (3), Lepidoptera (2), Blattodea (2) and 1 species each from Coleoptera, Odonata, and Mantodea. Diversity indices such as Shannon–Wiener, Simpson dominance, and Margalef indices were computed. Results of the study show that edible insect diversity has significantly decreased in the forest habitat. For a region highly dominated by entomophagy, such decreasing diversity raises a red flag. The field investigation showed that edible insects are highly sought after by local people. We identified the entomophagy practicing population mainly belonging to the Adivashi, Bodo, Rabha, and Sarania communities. They consume insects via different modes of preparation, such as fried, smoked, raw, etc. Moreover, people preferring entomophagy mainly belong to the youth (20–40 year) population. Therefore, our results conclude that MNP is a place vibrant with a high diversity, and abundance of edible insects. Further, it was found that these insects are good sources of protein, lipid, essential amino acids, omega-3, and omega-6 content, besides calcium, magnesium, and carbohydrate content. This validates edible insects as a future alternative source for an adequately nutrient-rich diet, proving to be majorly desirable in the context of food security. Preservation of such diversity necessitates the adoption of efficient and unique conservation techniques along with appropriate policymaking which can go a long way in augmenting greater insect diversity and also the food security of people in South Asia.

## ACKNOWLEDGEMENTS

We would like to thank Mr. A.M. Singh, PCCF, Government of Assam, and the Director of Manas National Park for their incredible assistance and support throughout the study. We also thank the Director, IAAST, Guwahati for his encouragement throughout the study. Special thanks to Prof. Scott V. Edwards, Curator of Ornithology in the Museum of Comparative Zoology, Harvard University for his valuable guidance. We express our sincere gratitude to the Swagat Health and Educational Trust for supporting this study through its entirety. We also thank the respondents and forest rangers/personnel associated with Manas National Park, for helping us in specimen collection and providing relevant

valuable inputs. Lastly, heartfelt thanks go out to the Centre for Environment, Education and Economic Development (CEEED) for giving us excellent logistic support.

### Funding
The authors received no funding for this work.

### Competing Interests
The authors declare there are no competing interests.

### Author Contributions
- Arup Kumar Hazarika conceived and designed the experiments, authored or reviewed drafts of the paper, and approved the final draft.
- Unmilan Kalita conceived and designed the experiments, performed the experiments, analyzed the data, prepared figures and/or tables, authored or reviewed drafts of the paper, and approved the final draft.
- Subhash Khanna conceived and designed the experiments, prepared figures and/or tables, and approved the final draft.
- Tarali Kalita performed the experiments, prepared figures and/or tables, and approved the final draft.
- Sangeeta Choudhury analyzed the data, authored or reviewed drafts of the paper, and approved the final draft.

### Field Study Permissions
The following information was supplied relating to field study approvals (i.e., approving body and any reference numbers):

Field study was approved by the Office of the Principal Chief Conservator of Forests (Wildlife) and the Chief Wildlife Warden, Government of Assam, India (No. WL/FG31/ResearchStudyPermission/19th Meeting/2019).

### Data Availability
The raw data are available as a Supplemental File.

### Supplemental Information
Supplemental information for this article can be found online at http://dx.doi.org/10.7717/peerj.10248#supplemental-information.

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
