# Peer review of "Diversity of edible insects in a Natural World Heritage Site of India: entomophagy attitudes and implications for food security in the region"

_PeerJ, doi:10.7717/peerj.10248_

## Round 0.1 · original submission · Major Revisions

Dear Dr. Hazarika and colleagues:

Thanks for submitting your manuscript to PeerJ. I have now received three independent reviews of your work, and as you will see, one reviewer recommended rejection, while another suggested a major revision. I am affording you the option of revising your manuscript according to all three reviews but understand that your resubmission may be sent to at least one new reviewer for a fresh assessment (unless the reviewer recommending rejection is willing to re-review).

In general, the reviewers wish to see improvements to English and grammar, as well as a better presentation of your findings (avoid duplicated results, cull unnecessary figures and tables, etc.).

The methods should be clear, concise and repeatable. Please ensure this, and make sure all relevant information and references are provided.

There are many minor suggestions to improve the manuscript. Note that reviewer 3 kindly provided a marked-up version of your manuscript.

Good luck with your revision,

-joe

·

Basic reporting

The language used is clear and unambiguous. The discussion was conveyed appropriately.
The literature review was sufficient. However, a few related studies could have been reviewed.
The structure is methodologically appropriate. The biochemical analysis added to the article’s sheen.
Figures are good. However, figures 10-16 seem unnecessary. These are already been represented in figure 17.
Raw data is robust and sufficient.
The biochemical analysis could have been restricted. Discussing diversity is more than enough.
Findings are significant and have good potential in defining the current edible insect scenario in Asia.

Experimental design

Research questions are appropriately defined.
Study area: Figure 1 is not mentioned. It should be mentioned.
Line 175: Table/diagram of the market price of some insects could be inserted (if possible)
Line
Lines 182-184: These could be removed.
Lines 185-186: The name of chemical compounds should be given.
The methodology otherwise is good. Figure 2 could be improved (if felt necessary).

Validity of the findings

The findings are worthwhile as they represent an area that has not been discussed before, mainly in the context of South Asia.

In the discussion part, the following studies could be cited:
Arya, M. K., & Verma, A. (2020). An Insight Into the Butterflies (Lepidoptera, Papilionoidea) Associated With Protected Area Network of Uttarakhand, Western Himalaya. In Current State and Future Impacts of Climate Change on Biodiversity (pp. 154-178). IGI Global.
Guadalquiver, D. M., Nuñeza, O. M., & Dupo, A. L. (2019). Species Diversity of Lepidoptera in Mimbilisan Protected‎ Landscape, Misamis Oriental, Philippines. Entomology and Applied Science Letters, 6(3), 33-47.
Dewan, S., Chettri, I. K., Sharma, K., & Acharya, B. K. (2019). Kitam Bird Sanctuary, the only low elevation protected area of Sikkim: A conservation hotspot for butterflies in the Eastern Himalaya. Journal of Asia-Pacific Entomology, 22(2), 575-583.\
Hagstrum, D. W., & Phillips, T. W. (2017). Evolution of stored-product entomology: protecting the world food supply. Annual review of entomology, 62, 379-397.

Supporting data is sound and appropriate. Figures are self-explanatory and support the conclusions.

Conclusions are appropriately stated. Limitations and future research prospects may be stated in the conclusion.

Additional comments

Overall, the study is an interesting one as it deals with lesser known facts of third world entomology. It could have been expanded to cover a broader perspective. However, the conclusions could be made clearer to reflect the findings. The mentioned minor revisions may be done.

·

Basic reporting

Needs English revision.
Text needs to be condensed; needs to be much more succinct.
Platitudes need to be avoided, repetitiveness needs to be reduced.
Important references are missing.
Figures are deficient,
Tables unclear without legends and units mentioned.
Methods re chemical analyses need to be described ( at the momentwhaat is written there is totally unacceptable!)

This ms needs a very, very MAJOR revision. It does, however, have potential and could be a useful contribution if revised adequately.

Experimental design

Major revision (especially regarding Chemical Analyses) is essential !

Validity of the findings

Avoid platitudes and follow my suggestions given in the Comments to the Authors

Additional comments

Ref Report 299

Entomophagy, edible insect diversity, and desirability of food security in a natural World Heritage Site of southern Asia
By Arup Kumar Hazarika et al.

This is a potentially interesting and comprehensive study and the authors ought to be encouraged to re-submit their work following a thorough revision. Although generally quite well written, there are several problems with this manuscript. In places it is very verbose and the text must be condensed; the methodology (especially with regard to the chemical analyses) is deficient; some important earlier publications have been missed; legends to tables and figures are either absent or deficient; some figures are deficient.
Title: to mention “…a Natural World Heritage Site of Southern Asia” is far too vague. Where in southern Asia? Southern Asia covers a huge area. Even later in the description of the study area, no information about where this “natural World Heritage Site” is located is given. Is it in Assam? Part of it in Bhutan? If so, that should be added. In Fig. 1 there should be a map between that overview map of India and that detailed view of the heritage site. The reader MUST have a map of the part of Assam (if the heritage site is in Assam) that shows the area. Also, maps must have a scale (x km).
The mentioning of desirability of food security in a Natural World Heritage Site is somewhat odd: food security is desirable only in the World Heritage site? Elsewhere it is not desirable?
Abstract:
Line 24: write “Insects not only play a significant role in the ecological processes of Nature, but since pre-historic times they have also formed a part of the human diet. With a still growing…”
Line 36: Isoptera have for several years now not been treated as a separate order of insects any more, but are included in the order of Blattodea.
Line 38: write “Biochemical analyses” (plural of analysis is analyses!). Further, what does “these” refer to in ‘these insects’ (no specific insets have been mentioned).
Line39/40: delete the lines “Further, the economic significance….It was observed that” and start “To manage insects…advantages.”

Introduction:
Line 49: the reference Yi etal. 2012 is misleading as Yi et al have not researched insect nutrients. There is no need for a reference at all with regard to the statement made on line 49.
L. 54: Likewise, Van Huis 2013 has never studied the diets of prehistoric humans and the reference this reference is simply wrong and either has to be removed or replaced by an appropriate reference.
Lines 60-62: the authors had better write: “Given the shortfalls of the ‘green revolution’ and high risk of food insecurity in developing and underdeveloped nations, the use of insects as a potential source of food for the burgeoning human population had been advocated by Meyer-Rochow (1975), a suggestion that has been gaining interest among researchers, entrepreneurs and policy makers worldwide ever since.”
Line 63: replace “in most Asian…” with “ in some Asian…”
Line 64: write “Weaver ants (Oecophylla smaragdina), whose chemical composition and value as a human food item has been assessed by Chakravorty et al. (2016), are widespread …”
L67: Delete “Besides” and start “In Thailand over…”
L70, 71, 72: Style ! What do the words “their cultural…”, “But they…”, “They are…” refer to?
L73/74: provide scientific name for the ‘red ants’ and clarify “…their eggs”. Almost certainly NOT eggs (the eggs are miniscule, more likely their pupae!)
L76/78: Condense and make one sentence. Drop Borgohain et al. 2014 ,as it is the wrong reference.
L79/80: the reference to “various ethnic tribes” is too vague. There are approx. 50 ethnic groups in Arunachal! Better use Chakravorty et al. 2013 !
L83: write “Odonata were consumed…”
L86: start the paragraph with “Scientific reports…”
L89/90: Why mention aquatic beetles in Latin America and Africa? Macadam & Stockan 2017 is good reference, but why not cite Indian research on edible beetles like Shantibala T, Lokeshwari RK, Debaraj H 2014 Nutritional and antinutritional composition of five species of aquatic edible insects consumed in Manipur. J. Insect Sci. 14 (4), 1-10
Or: Ayekpam, N. Singh NI, Singh TK 2014 Edible and medicinal insects of Manipur. Ind. J. Entomology 76 (3), 256-259
Or Asian research on edible beetles: Ümit İncekara and Hasan Türkez 2009. THE GENOTOXIC EFFECTS OF SOME EDIBLE INSECTS ON HUMAN WHOLE BLOOD CULTURES.Mun. Ent. Zool. Vol. 4, (2), 531-535 (this article deals with 3 species of edible aquatic beetles (Hydrophilus, Cybister and Dytiscus)
Or: Jäch MA 2003 Fried water beetles Cantonese style. Am. Entomol. 49, 134-137.
L99: Write “Information on the nutrient composition (macro- and micronutrients) of the edible insets species was used to explore …”
L101: Write “In short, the aims of this study have been to determine the degree to which the ethnic people use insects in their diet and which species they consume. Recording seasonal abundance and availability of edible species as well as evaluating the role that entomophagy could possibly play as a measure of food security in the region, were further aspects of this study.”

Materials and Methods
Describe clearly where the national park is located. Add a proper map (with scale)
L124: “these tribes…”? Which? Has Rabha 2016 and have Das & Hazarika 2019 studied all of them? They (who?) really rear insects??? Which species (there are several lakh of insect species in India?

Sampling: more details are required. Describe the equipment used for sampling. How often were traps emptied? Who was involved in the sampling?
Figure 2 does not give ANY information on traps, the frequency that collecting took place, people involved, whether collecting occurred during the day or nighttime hours, if light traps were used (and if not why not), what kinds of traps for aquatic insects were used?

L140-146: approximately what were the total areas in sqkm that the 4 different habitat types covered?

L172: “…once per season….and twice during the whole survey duration” is not clear. Explain what seasons you distinguish in that area (perhaps spring, summer, autumn, winter? Or maybe monsoon and pre-monsoon; rainy and dry season?), but which months are involved. The project was a one-year project: = 4 seasons? But “twice visited per season”? Not clear “during the survey duration”, but ”once”?

More information on the respondents is required: how were they chosen? Men/women children? Ages involved and if possible some background of the respondents (literate? Farmers? Housewives?)

L180-195: This is one of the least satisfactory parts of the manuscript. You used fresh or dried insets? How were they dried? Values given are per wet weight or dry weight. You write that for this study “we chose insect species of the order Coleoptera” but Table 5 lists species belonging to all kinds of insect orders!
You need to provide proper references for standard methods, e.g. Bragdon’s method, etc. It sounds as if for protein analysis the Lowry method was used. But there is a chance for overestimation as some flavonoids, polyphenols also contribute in the development of the colour. The Micro-Kjeldahl process is superior and more reliable to estimate N and then protein.
What about the chloroform-methanol solution (ratio, solvent???). What about the Nelson method: provide reference or describe then method in detail.
Tables, e.g. 5: nutrient composition, what are the units? How to prepare the sample for analysis, e.g., then sampling process?
Which method(s) was (were) followed to determine omega-3 and omega-6 fatty acids? What is the unit? Hpw were mineral compositions determined? Where are the statistics ? How many replicates were done?

You should look at Sampat Ghosh’s detailed descriptions of the methodology to see what info is required:
Journal of Asia-Pacific Entomology 19 (2016) 487–495 “Nutritional value and chemical composition of larvae, pupae, and adults of worker honey bee, Apis mellifera ligustica as a sustainable food source”
Sampat Ghosh, Chuleui Jung, Victor Benno Meyer-Rochow
Materials and methods
Sample collection and preparation
Specimens of A. mellifera ligustica were collected from the experimental apiary of Andong National University (36°34′N; 128°43′E) during June, 2015. The movable frame combs were taken to the insect lab
and the bee brood (consisting of 4th and 5th instar larvae prior to cell capping as well as pupae 2–5 days after cell capping) was removed from the comb. Adults were collected from the inside of the hive during
the day and consisted almost entirely of ‘house-workers’ (not more than 3 weeks old after emergence) that had not been actively foraging outside. The developmental stages i.e. larvae, pupae and adults, were separated in the laboratory, wings of the adults were removed, then oven-dried, ground to powder and prepared as dry matter (DM) for further analysis. All the solvents and chemicals used in the study were of analytical grade.
Proximate analyses
Proximate compositions, i.e. moisture content, crude protein, crude fat, crude fibre, ash and nitrogen free extract (NFE) were estimated following standard methods recommended by the Association of Official Analytical Chemists (AOAC, 1990). Moisture percentage was calculated by drying the sample in an oven at 100 °C for 2 h. The dried sample was put into desiccators and allowed to cool and reweighed. The process was repeated until constant weight was obtained. Crude protein was determined by the Kjeldahl method and total protein content was calculated as the amount of total N determined
multiplied by nitrogen-to-protein conversion factor of 6.25. Fat percentage was calculated by drying fats after extraction in a Soxhlet using Diethyl ether. Ash percentage was calculated by combusting the samples in silica crucible placed in a muffle furnace.
The percentage of carbohydrate was determined by subtracting all of the components (crude protein, crude lipid and ash) from 100. The calorific value (kcal/100 g) was computed by multiplying the
factors for carbohydrate and protein by 4 each and that of fat by 9 and then taking the sum of the products. All of the analyses were performed in triplicate and expressed as mean ± standard deviation.
Amino acid analysis
Amino acid composition was determined by Sykam Amino Acid analyser S433 (Sykam GmbH,Germany) following the standard method of AOAC (1990). The ground samples were hydrolyzed in 6 N HCl
for 24 h at 110 °C under nitrogen atmosphere and then concentrated with rota-evaporator. The concentrated samples were reconstituted with sample dilution buffer supplied by the manufacturer (0.12 N, pH 2.20). The hydrolyzed samples were analysed for amino acid composition. The operating condition of the amino acid analyser was as the following: Instrument Amino acid analyser (Sykam GmbH, Germany) Column LCA K07/Li (PEEK — column 4.6 × 150 mm), Application Physiological Detector Photometer (570 nm, 440 nm), Detection principle Ninhydrin reaction, Inert gas N2.
The amino acid score was calculated based on FAO/WHO/UNU (2007).
Fatty acid analysis
Fatty acid composition was analysed by gas chromatography flame ionization detection (GC-14B, Shimadzu, Tokyo, Japan), following the standard method of the Korean Food Standard Codex (2010). The
samples were derivatised into fatty methyl esters (FAMEs) following the method described by Lepage and Roy (1986). Identification and quantification of FAMEswere accomplished by comparing the retention times of peaks with those of pure standards purchased from Sigma and analysed under the same conditions. The results were expressed as a percentage of individual fatty acids in the lipid fraction. The operating condition of GC for fatty acid analysis was as the following:
Instrument Gas Chromatography (GC-14B, Shimazdu) Column SP-2560, Detector Flame Ionization detector (FID), Carrier gas N2, 300 KPa, Initial temperature 170 °C, Initial time 0 min, Programme rate 1 °C/min, Final temperature 205 °C, Final time 5 min, Split ratio 100:1.
Minerals analyses
Minerals were analysed following the standard method of the Korean Food Standard Codex (2010). The dried powder samples were digested with nitric and hydrochloric acid (1:3) at 200 °C for 30 min.
Each sample was filtered using Whatman filter paper (0.45 μm) and stored in washed glass vials before analysis. Minerals were analysed by inductively-coupled plasma-optical emission spectrophotometer
(ICP-OES 720 series Agilent).

Or look at:
Journal of Asia-Pacific Entomology 20 (2017) 686–694
Nutritional composition of five commercial edible insects in South Korea
Sampat Ghosha, So-Min Leeb, Chuleui Jungb, V.B. Meyer-Rochow

Results (FIGURS AND TABLES):
The Tables 1 -4 are very good and detailed. Congratulation, well done!
BUT: why is there no Oecophylla smaragdina Table 2 ?
For Table 4 you need to explain in a legend: Occurrence, FBH ,OFH, Abundance and Relative Abundance.
Figures 1, 2, and 3 need to be improved!
Fig. 3 looks nice, but you have to give the scientific names of the insects you depict in the figure !
Fig. 4 is OK.
Fig. 5 is terrible: What is the y-axis (number of peopåle interviewed, percentage of population?) What is the x-axis (distance in km?). How can the surface of the data be ‘curved’ if you have separate data points . Explain.
Fig. 6: shows men or women?
Figures 7 and 8: good
Figures 9 – 17: inadequate. What is the unit for the data? Based on dry or wet weight? You need to give the scientific names of the insects studied. Based on what quantities (numbers of insects, weight used for analysis; how many replicates?)


L210: “The order Hemiptera….” “The order Coleoptera…)
L212: “A total of 9,213 edible insects…
L213: …the field observation.”
L220: “Species of the order…”
L226: “…the year, but in the winter they were less abundant than during the pre-monsoon and monsoon season.” Which months is winter?

L231: This paper should be cited:
Journal of Asia-Pacific Entomology 17 (2014) 407–415
Nutritional composition of Chondracris rosea and Brachytrupes orientalis:
“Two common insects used as food by tribes of Arunachal Pradesh, India”
Jharna Chakravorty, Sampat Ghosh, Chuleui Jung, V.B. Meyer-Rochow

Discussion
L271- 330: This ecological section is useful and interesting and should be retained, even though it is not directly (but certainly indirectly) related to edible insects of the region.
L277: scientific names of insects should be italicized.
L296: “This adverse finding…”
L330-334: “For instance,….their lives” should be deleted
L340: “…of the Bodos…”
L346: write “…is from the order Orthoptera, which comprise 8 species of which 7 are short-horned …”
Or you can write “…, Orthoptera, which contain 8 species…”
L348-353: Delete “Notably, Assam…and so on.”
L358: “…are collected (cf., Meyer-Rochow 2017). This indicates…”
L362: “…rather than an…”
L369: Vaccaro et al. (2019): reference is incomplete!
Add: “…observation of Vaccaro et al. (2019), but does not agree with the study by Ghosh et al. (2020), carried out in Ethiopia. (Journal of Insects as Food and Feed 2020, 6 (1), 59-64).
L376: wrong general statement: not ALL insets contain minimal amounts of carbohydrates. No, some like bees, honeypot ants etc are VERY rich in carbohydrates!
L377: How can you cite a 2012 paper and then write that the results were “further verified” by a 2006 paper!!! The citations on nutrients L377, L379, should be replaced by newer and more reliable ones.
L384: write “…insects should potentially be able to supplement the diet of livestock.”
L391/2: Payne et al 2016 noted that there is NO SIGNIFICANT DIFFERENCE (!) in the nutritional qualities between insects and conventional protein food sources like pork, beef and poultry! Get this right, please.
L396: benefits: like what?
L405408: Delete “For instance….areas.”
L420: “…ladybird beetles..,”
L425: yes, here you mention it “rich in protein and carbohydrates” but earlier on Line 376 you wrote carbohydrates were minimal!
L434: Delete “be”
L436: which larvae? And who found that (reference please)
L 440: “…higher than that of the adult…”
L445 – 458: Delete the whole paragraph.
L467: you should add after “security “…like learning from other countries like, for instance, South Korea (Meyer-Rochow V.B., Ghosh, S., Jung, C 2019 Farming of insects for food and feed in South Korea: tradition and innovation. Berliner und Münchener Tierärztliche Wochenschrift 131 (5/6), 236-244.
L469: Write “Our study shows that edible insects are of considerable nutritional value and expanding their acceptability as human food can be expected to improve the nutritional status of people and possible reduce their costs.
L472: you mean ‘six’, but you need a reference to back up that statement!
L478:”Certain insects…” ? Which? Silkworms, honey bees, and as of late bumble bees and wasps. Is Akerele et al 2018 a correct reference?
L480: how can you leave out Korea with hundreds of years of insect cultures!
L482: Specht et al. 2019: the title is missing in the references!
L486: the statement “ Insect diversity…. 2018).” Is it relevant for Assam and the Heritage Site?
L488: “These activities…” (WHICH?) “…improving their diets… (WHOSE?) and their livelihoods (WHOSE?)
In the 4 lines 486-489 you use the word “also” three times
Conclusion: Make sure you do not simply repeat what is already in the Abstract.
L513: why ‘may be concluded’? You are doubting your own results?
L516: whose ‘status’?
L518: why ‘also’. Delete ‘immense’. We have NO IDEA if the economic advantages will be or “are immense”.
Rewrite this whole Conclusion in one short paragraph. Go over the entire paper and see if you can streamline and condense the text! Avoid platitudinous statements.

Reviewer 3 ·

Basic reporting

In general, the experiments are properly planned and executed but the data presentation is extremely poor with many figures duplicating the findings and the entire manuscript is not straightforward. There are several instances in entire manuscript where English grammar and syntax needs great improvement and several sentences are very difficult to follow. Overall, the writing needs more work. For details, please see specific comments below.

Experimental design

The Materials and Methods section in this paper would require additional details for clarity. It is quite shallow and does not clearly state out the research design. I think the whole M &M section needs to be rephrased in simple and proper English grammar. It is a bit difficult to follow because of so many errors.

Validity of the findings

The result section is also poorly structured and not straight forward to follow as it lacks coherence in interpretation of findings because of lots of grammatical errors. This section should also be carefully rewritten with focus on key findings.

Additional comments

TITLE PAGE:
The title of the paper needs to be carefully revised because it is not very clear and does represent the context of the paper. The title too wordy. A possible title could be “Composition and nutritional profile of edible insects in the Manas National Park, India: Implications for food security in the region”
This paper also lacks a running heading, which is one of the key instructions to authors

ABSTRACT:
The abstract is too wordy and does not state the problem and the aim of the study clearly. The results should be presented in a systematic fashion and the authors should concentrate on trends while maintaining the number of word counts per the journal format (Background, Methods, Results and Discussion). I think this section should be re-written altogether in good English for clarity. Also, the nutritional information is completely lacking in the abstract.

INTRODUCTION
This section does not situate the exact importance of this study and it is full of grammatical errors. Most of the paragraphs will need to be revised and rephrase accordingly. Most of the sentences are very confusing the objective of the work is well explained.

DISCUSSION
The discussion has a very poor scientific presentation format. However, I think the discussion section needs to be rewritten after proper presentation of results. Authors should avoid presenting results in this section. The authors should seek the assistance of an English speaker to address the numerous errors found in this section. Very poor conclusion drawn from the work.

ACKNOWLEDGMENTS: No comments.

REFERENCES
The references have failed to follow the format of the journal

FIGURE CAPTION
No figure caption mentioned in the text.

FIGURE
Most of the graphs are poor done and the some if the axes are overcrowded. Figures have been presented with no labels. Figure 9 is not reported in the Result section. However, I think these figures should be presented in a tabular format. Figure 10, is not necessary because it is a summary of all the figure 9s. Again the different elements (minerals) were not measured using the same approach as such the units of measurements are different. Therefore, they can be lump together as shown in figure 10. I think figure 10 should be deleted.


TABLE
The abbreviations in Table 3, should be written in full as a footnote below the table. The headings of the tables should not be centralized.

Annotated reviews are not available for download in order to protect the identity of reviewers who chose to remain anonymous.

---

## Round 0.2 · Major Revisions

Dear Dr. Hazarika and colleagues:

Thanks for resubmitting your manuscript. One review is positive and recommends acceptance; however, another has raised more concerns. Please look over these concerns raised by reviewer 2 and reviser your work accordingly.

Your manuscript seems to contain several inaccuracies, ambiguities and false statements. Also, there are many spelling mistakes and other grammatical issues. Please enlist the help of an English expert.

Of more concern, please address the problems raised by reviewer 2 with your chemical analyses! Some figures and Tables appear to contain many errors and must be fixed (or removed).

It might be that your manuscript will be in more publishable form if you restrict the work to the ecological findings and conclusions regarding insect uses as food. Please consider this. However, please carefully revise your work and cull inconsistencies (e.g., you report one species of edible Coleoptera in the Abstract and most of the text, but then discuss two species later).

I look forward to seeing your revision, and thanks again for submitting your work to PeerJ.

Good luck with your revision,

-joe

·

Basic reporting

All recommended changes have been appropriately done.

Experimental design

The experimental design has been improved. I accept the changes and do not recommend any further changes.

Validity of the findings

I accept the revisions done.

Additional comments

I accept the revisions made and recommend the paper for publication.

·

Basic reporting

Manuscript is full of inaccuracies, false data, contradictions, unacceptable data of the chemical analyses.

Experimental design

Not entirely clear.
Chemical analyses are not fully explained.
Results more than dubious; some are totally wrong.

Validity of the findings

Ecological part ok;
List of insects with many errors (spelling and contradictions).
Abstract and ms content in disagreement. Text and some Figures/Tables likewise.

Additional comments

Ref Report 305

Composition and nutritional profile of edible insects in a Natural World Heritage Site of India: Implications for food security in the region
By Arup Kumar Hazarika et al.
The authors have submitted a somewhat improved version of their manuscript, but the modifications were largely cosmetic and serious scientific problems were not addressed. The authors did not take their time to spot and correct inaccuracies and contradictions, which this manuscript is full of. Some of the figures and tables are still unacceptable and if the authors cannot get the ms into an acceptable shape, it will have to be rejected, which would be a pity as there are some useful findings and suggestions in this ms. I shall now go through the ms page-by-page and line-by-line.
I wsh to let the authors know NOT to be disheartened by my comments and suggestions: they are meant to help the authors to turn their results and their ms into a ‘product’ theycan be proud of; a publication that will stand the test of time and will be cited by other researchers. But to get the paper into the required ‘shape’ takes more than just a few cosmetic changes. The authors need to spend time and carefully read and re-read the ms many times to iron out inconsistencies, to streamline the text, to remove ambiguities and inaccuracies, correct spelling mistakes etc. Do not rush! Take your time and do a GOOD job revising the ms, because otherwise you’ll end up with a rejection.
Abstract
L29: delete ‘intend to’
L30: already criticized the first time, write “…located in Assam (India).”
L34: not nine orders, but eight ! The next sentence actually lists them.
L39: “…were carried out to record….their possible role as nutrient inputs.”
L41: delete ‘an’ and write “…sustainable collecting and rearing methods, emphasizing …”

Introduction
L52: “As such, they arouse interest not only…”
L53: “ A dimension of their existence not to be overlooked pertains to…”
L55: “…insects form or formed a part of…”
L57: “…is nowadays no longer a traditional or common …, except for some…”
L64: many Asian nations…
L67 “…found from China’s South to northern Australia…”
L68: “Although edible insects are not yet of much commercial value, some economic and…are available (Johnson 2010)” NOTE: 2010 is 10 years ago! Hasn’t the situation changed since then?
L70: delete ‘by its people’. “Approximately 50 insect species are consumed in the north and about 14 species in the southern part of Thailand…”
L72: delete ‘may’
L73: “…religion and place they call home.”
L75-77: Delete, because that sentence is totally out of place with no connection to previous or following sentences.
L82: delete ‘of Arunachal Pradesh’
L85: write “…by the Galo and Nyishi tribes.”
LL89: NOTE: only proteins and vitamins? Not also lipids and minerals and perhaps fibre and carbohydrates?

Material & Methods
L124: Temperatures range from…
L137: Sweep nets were used for collecting grasshoppers…
L138: naturally hide in low grass NOTE: why ‘naturally’? Can they also hide unnaturally?
L139¨netting was normally carried out during daytime, BUT see Line 135, where it says during the early hours 0500-0900 !
L143: deleted ‘for a long time’. NOTE: Provide characteristics of the CFL bulb, explain how far from the white cloth; explain size of the white cloth. If the light was BEHIND a white cloth, why was the bucket of water not in front of the white cloth. At what time of night and for how long were insects trapped in this way. When and how often were such collections made. Readers should know the seasons when such collecting method was used.
L154: NOTE: you write grasshoppers and beetles were collected by hand, but on Line 137 it says grasshoppers were collected by sweep netting !
L158-159: explain or use references to explain the terms ‘Jakoi’ and ‘Saloni’.
NOTE: what kinds of waterbodies were collections made from and when and how often did that take place? E.g., rivers, swamps, brooks, ponds?
L169: are alcohol and formaldehyde not “standard methods”?
L171: write “…Shillong, Meghalaya (India).
L185: is it ‘Borthakur’ or ‘Barthakur’ as in the references?
L188: delete ‘also’ NOTE: these ecological indices are useful. I like them.
L216: 2672 respondents from 30 villages, BUT what would interest the reader is the number of respondents per tribe studied!
Biochemical analyses (use the plural ‘analyses’ not singular ’analysis’. NOTE: this goes for ALL the analyses: how many replicates were carried out! From which part of the insect body was the tissue taken: muscle tissue, fatbody, gonads, brain???
L223: do you really mean just ‘estimated’ or do you mean ‘determined’?
NOTE: where is the detailed description on how you obtained data on the various amino acids and their amounts?
L236: “…was done according to the …”
L245: NOTE: for lipid analyses how were omega-3 and omega-6 fatty acid amounts obtained?
L254: How the specimens were prepared for mineral analyses is not explained properly.

Results
L265: the order Isoptera no longer exists; termites are now with Blattodea! Correct also Table 2 and 4.
L271: Table 2, table 4, figure 4 and also the text of the ms contains spelling errors. Please correct: Mecopoda elongata, Hieroglyphus banian, Acheta domesticus, Antheraea assama, Oryctes rhinoceros !
L278: the beetle Hydrophilus olivaceus, mentioned here, occurs in no table or figure! Instead, it says in several places that only one species of beetle (Oryctes rhinoceros) was considered edible (and probably only the grubs and not the adult, but that seems to go unmentioned).
L287: Hieroglyphus banian
L296: “…the order Orthoptera…”
L295: “…followed by Coleoptera (8.02% while the Odonata has the least relative abundance” NOTE: this cannot be right as you have only one species of beetle. Explain.
FIGURE 4 is inadequate: what about the x-axis? There needs to be a scale or some numbers. Furthermore you write Mantis religiosa is most abundant in the monsoon season, but Fig 4 suggests that it is Periplaneta! In winter Vespa affinis has the highest ‘availability’ (really? Isn’t Periplaneta equally abundant?)
L294-304: the whole paragraph needs to be rewritten; Figure 4 needs some mumbers (do not give numbers in brackets in the text, but enher them in the text!). Correct the spelling mistakes.
Explain what “highest insect abundance during monsoon season (4808)” means and how that can be gleaned from Figure 4 !’
Paragraph L306 – L321 is equally problematic. The ethnic communities studied are mentioned in Fig 5. But that figure is useless without an explanation of how many respondents were there for the various ethnic groups. You need a vertical scale ! And what does ‘yellow line’ mean? There is no yellow line, unless you mean the ‘yellow column’.
Figure 6: meaningless unless you explain which tribe this refers to and what the little numbers in the columns mean. For ALL the figures and tables you have to have a brief text, a ‘figure legend ’to explain what the figure or Table contains.
The results, nicely summarized in Fig. 7: but for which tribal group is this applicable? Likewise Fig 8 (in which it surprises me that chutney/pickles do not feature! No insect pickles?)
NOTE: In figures 9 and 10 you are using ‘common names’ like house cricket, crickets, eri and even ‘water beetle’ (!), which did not appear at all until then, not even in Table 2. Please check ALL of your figures and Tables.
BE CONSISTENT ! Do NOT use ‘common names’ in some tables and scientific names in others, but stick to the scientific names in all tables and figures !
Figures 9 and 10 are wholly inadequate. You can’t use common names here (like grasshoppers, small, brown, large, etc) when in Table 2 and Fig 4 you use scientific terms! In 9 A – E the sequence of the insects is the same, but in F it is different: why?
You need an explanatory legend to say that the values entered are based on analyses of 100 mg. Even so, the data are more than dubious: magnesium content is as high as carbohydrate (see lengths of the bars!): it can’t be! Another example: for mole cricket lipid you state 0.055, BUT omega-6 content is 0.24781: impossible! (I earlier mentioned that you did not provide information on how you measured omega-3 and omega-6 fatty acids!
You do not mention on how many replicates the data are based!
You are listing figures with up to 5 decimals behind the comma, which is ridiculous: you are dealing with mg !!! Using with 5 digits behind the comma you are in nanogram range!
I feel you ought to ditch (= delete all the data in Fig 9 as they are unreliable, which cannot be good for you and your institute’s reputation.
Likewise Figure 10 (incidentally, suddenly there is a water beetle again in addition to the rhinoceros beetle!): the data are all wrong unless fully explained in a figure legend how you constructed this figure. According to the figure, which lacks units and numbers, calcium and magnesium content (taken together) is higher than lipid !? That whole figure should also be ditched as it is unreliable.
By the way, do you say anywhere in the text which amino acids you refer to as ‘essential’ and how you determined amino acids qualitatively and quantitatively?

Discussion
First paragraph, spelling: Mecopoda elongata, Hieroglyphus banian, Acheta domesticus, Gryllotalpa africana. “and so on” does not sound very elegant; perhaps you could write “and some others”.
L333-334: “…insects are mainly those of the Adivashis… Sarania. A section…”
L 391-397: you reported 22 edible species, of which one species was a Coleoptera. How can you now wite that 16.66% were Coleoptera? In fact, with such a small number of species (22), it is not necessary to express numbers in percentages; it’s useless. Use real numbers and NOT percentages. percentages are only useful if you have large numbers to deal with.
L414: “…does not agree…” is misleading. You need to write “…(2019), but in Ethiopia young people are less inclined to eat or even taste insects (Ghosh et al. 2020).”
L420-421: you write “…are very rich in carbohydrates. Our study verifies this fact as we can see from Figure 7 that the insects are rich in protein, but have minimal carbohydrates.” You just wrote in the first half of the sentence “rich in carbohydrates’ and now it’s “minimal carbohydrates’. That is another one of those contradictions in your ms. Besides, you refer to Fig. 7, but Fig. 7 shows “Reasons for practicing entomophagy”!
L427: you write about a “moderate presence of magnesium”, but on Line 319 you write magnesium content is minimal!
L449: you need a reference here to back up your statement!
L451: why “also”? In fact, there are many places you use “also” and frequently it is not clear why ‘also’. For example ‘also’ on Line 477 and many others.
459-473: that is a terrible paragraph, muddled, inconsistent and about beetles that are either totally inedible (like ladybirds) or are not mentioned in this ms, because you made the earlier statement (n the abstract as well) that there was only one edible coleoptera species, namely Oryctes rhinoceros. I cannot even start correcting this paragraph. Unless completely rewritten, the paragraph should be dropped.
L478: lipid content of larvae… Which larvae?
L481: you do not examine fat content of pupae and larvae of edible Coleoptera of your one species of beetle. If you want to use a reference to show differences in fat (and other contents) between larvae, pupae and adults use this reference: Ghosh et al.2016 “Nutritional value and chemical composition of larvae, pupae, and adults of worker honey bee, Apis mellifera ligustica as a sustainable food source” Journal of Asia-Pacific Entomology 19 (2016) 487–495.
L487-500: reduce this paragraph to three sentences.
L502/3: Write “Rearing insects can result in environmental benefits.... Insects can impact organic farming and help to reduce…emit fewer greenhouse gases….compared with conventional livestock…”
L512: Our study confirms…
L514”…possibly reduce the insects’ costs. ….With a wider insect diversity, ….of people should improve while….
L516: ‘fish’? Which fish? There are more than 40,000 species of fish with very variable fatty acid contents.
L523: “…domesticated since they are of high…”
L525: delete ‘immensely’
L526 ?...help exploiting new…
L530-537: cite Gahukar, R.T. Edible insects collected from forests for family livelihood and wellness of rural communities - a review. Global Food Security 2020, Doi.org 10.1016/j.gfs.2020.100348.

Conclusions
L541: Write “In this study we recorded edible insect diversity…”
L544: not nine, but eight orders of insects.

References:
I did not check the references for accuracy and leave that to the authors.

Figures and Tables
Already commented on in detail (see above text): all need to be modified and some should be dropped.
Figure 1: ok
Figures 2 and 3 are unnecessary. Fig. 4 needs to be improved (see earlier comments above); Gigs 5-8 should be improved (see earlier comments above)
Figures 9 and 10 contain inaccurate data and should be deleted.

Table 1: ok Table 2: needs to be corrected; Table 3: ok; Table 4: scient. Names to be corrected;
Table 5: completely false data (e.g. termite and cockroach calcium data are almost twice that of protein! Wasp larvae twice as much magnesium as lipid! All of these data are completely wrong. Delete this terrible table as quickly as possible!!!

---

## Round 0.3 · Minor Revisions

Dear Dr. Hazarika and colleagues:

Thanks for revising your manuscript. The one reviewer willing to re-review is mostly satisfied with your revision (as am I). Great! However, there are a few minor edits to make. Please address these ASAP so we may move towards acceptance of your work.

Please also seriously consider the reviewer’s comment about your title, as well as references missing from the original submission.

Best,

-joe

·

Basic reporting

Much better than before, but title needs to be re-worded and the reference list still contains dozens of publications from the earlier ms, which are now NOT any more in the text of the revised version. That must be corrected.

Experimental design

OK

Validity of the findings

OK

Additional comments

Ref Report 309
Title: Composition and nutritional profile of edible insects in a Natural World Heritage Site of India: Implications for food security in the region
By: Arup Kumar Hazarika et al.
The revised version of the manuscript is much improved, but some inaccuracies remain and need to be corrected. Also the title had better read:
“Diversity of edible insects in a Natural World Heritage Site of India: entomophagy attitudes and implications for food security in the region”
Having composition and nutritional profile in the title is misleading and sends the wrong message.

L38: write “Aspects of the economic significance of entomophagy were also observed during the field investigation.”

L55: insert after “…is referred to as entomophagy” the reference: Evans J, Alemu MH, Flore R, Frost MB, Halloran A, Jensen AB, Maciel-Vergara G, Meyer-Rochow VB, Münke-Svendsen C, Olsen SB, Payne C, Roos N, Rozin P, Tans HSG, Van Huis A, Vantomme P, Eilenberg J. ‘Entomophagy’: an evolving
terminology in need of review. J Insects Food Feed. 2015;1(4):293–305. Add this to the references list.

L68/69: the reference to Johnson 2010 is incomplete in the reference list. Page numbers are missing and the name of the book’s editors are Durst, P.B.; Johnson, D.V.; Leslie, R.N. and Shono, K. (eds.). The publisher is issing: FAP Publ., the place is missing: Bangkok.

L69: delete “In Thailand…eaten.” And write “Approximately 50 insect species are eaten in Thailand’s north and about 14 species are eaten by people in southern Thailand (Rattanapan, 2000).”

L 73: “…characteristics can also be important (Dumont, 1987).

L78-80: delete “In Arunachal….. Further” and write “A total of 81 species are eaten in Arunachal Pradesh by the Galo and Nyishi tribes (Chakravorty et al. 2011).” NOTE: this is not the reference you currently have in the list.

L90/91: write “…African nations (Macadam & Stockan, 2017) and are popular in Manipur (India) (Shantibala et al., 2014).”

L134: you write “…as it required good vision” Good vision by whom? Best drop that part of the sentence and continue “….as we could not collect nocturnal taxa in this way.”

L138/139: it makes no sense to have a light source behind a white cloth and then place a bucket under the light source! The bucket should be IN FRONT of the white cloth !

L146: replace ‘them’ with “it”

L150: “..or in the evening…”
L151: “Mole and field crickets were dug out…”
L162: replace ‘insects’ with “them”
L170: write: “…around MNP during the period 2018 (June)- 2019 (June). And delete “The sampling was done” at the end of Line 170.
L213/215; 265; Table2 (local name): Not clear why in the column titled “Local name” the Bodo language was chosen. The majority of the respondents were Adivashi not Bodo. Explain please.

L285: “…insects were mainly…”
L93 “Our results show….”
L306 “…has warmed by over 0.5C…” (you need a reference here for that statement!)
L32: The decreasing pattern…”
L 326/7: not clear “…with time while remaining constant at some point.” That seem contradictory!
L327: “This calls for urgent….” (What do you mean by “This”???)
L330-334: good point !
L 367: “The majority of…”
L378: write “….edible insects require higher energy in culture and contain higher sodium and saturated fat content…”
L389: “This frequently uproots the livelihood…”
L415: wrong citation Van Huis 2003 does not mention with even one word the situation in Laos. You want this reference: Meyer-Rochow VB, Nonaka K, Boulidam S 2018 “More Feared than Revered: Insects and their impact on human societies (with some specific data on the importance of entomophagy in a Laotian setting)” Entomologie Heute 20, 3-25.

L489: please delete this part of the sentence “and Prof V.B. Meyer-Rochow, Department of Ecology & Genetcs, University of Oulu, Finland”

References: Delete Chakravorty 2009 and delete reference 8, but replace reference 8 with: Chakravorty J, Ghosh, S. & Meyer-Rochow, V.B. 2011. Practices of entomophagy and entomotherapy by members of the Nyishi and Galo tribes, two ethnic groups of the state of Arunachal Pradesh (North-East India). Journal of Ethnobiology and Ethnomedicine 2011, 7:5

L644: delete reference 54.

What is really ANNOYING and I can not find excuses for is the lack of attention by the authors to their list of references! Numerous entries are NOT mentioned/cited in the text and therefore should NOT appear in the list of references.
The references that should be removed (or cited in the text) are
References 4, 6, 8, 13, 19, 27, 29, 38, 39, 41, 44, 47, 49, 53, 54, 55, 56.
An undergraduate tesis (ref 53) is not something that's citable, unless it is available on the web (in that case give webpage).

Figures and Tables now more or less ok!

---

## Round 0.4 · accepted · Accept

Dear Dr. Hazarika and colleagues:

Thanks for revising your manuscript based on the concerns raised by the reviewers. I now believe that your manuscript is suitable for publication. Congratulations! I look forward to seeing this work in print, and I anticipate it being an important resource for groups studying entomophagy. Thanks again for choosing PeerJ to publish such important work.

Best,

-joe